# Aligning Predictive Uncertainty with Clarification Questions in Grounded Dialog

**Kata Naszádi[1,2]** and **Putra Manggala[2]** and **Christof Monz[1,2]**

[1]Language Technology Lab
[2]University of Amsterdam

k.naszadi@uva.nl

## Abstract

Asking for clarification is fundamental to effective collaboration. An interactive artificial agent must know when to ask a human instructor for more information in order to ascertain their goals. Previous work bases the timing of questions on supervised models learned from interactions between humans. Instead of a supervised classification task, we wish to ground the need for questions in the acting agent's predictive uncertainty. In this work, we investigate if ambiguous linguistic instructions can be aligned with uncertainty in neural models. We train an agent using the T5 encoder-decoder architecture to solve the Minecraft Collaborative Building Task and identify uncertainty metrics that achieve better distributional separation between clear and ambiguous instructions. We further show that well-calibrated prediction probabilities benefit the detection of ambiguous instructions. Lastly, we provide a novel empirical analysis on the relationship between uncertainty and dialog history length and highlight an important property that poses a difficulty for detection.

## 1 Introduction

Human language offers a natural interface between machine and human when solving a collaborative task. We wish to build an artificial agent that can not only ground natural language in its environment, but can also solicit more information when communication fails or the human's instruction was ambiguous. Asking questions establishes two-way collaborative communication instead of silent following of instructions. Endowing an instruction-following agent with the ability to prompt questions poses two related but different problems: when to initiate asking a question and what the content of the question should be. In this work, we focus on the first problem.

The problem of deciding if a system should be asking clarification questions has been previously discussed in the context of conversational

search and dialog systems. Past approaches can be grouped into two paradigms, where a separate model decides if a follow-up is necessary (Kim et al., 2021; Song et al., 2009), and where the need for clarification is based on a model that would eventually generate the response (Arabzadeh et al., 2022). In the former, a separate training set is needed with annotations that indicate ambiguity (Aliannejadi et al., 2021). In the latter, no such signal is needed – the system uses an estimate of quality of its own predictive performance to decide.

The same problem can be posed in the context of visually grounded dialog. The domain of task-oriented communication in a visual environment is less mature, and the need for questions is often hard-coded in these systems (Das et al., 2017; Greco et al., 2022; Lee et al., 2018). Nevertheless, we can already find examples of both aforementioned paradigms, where a binary classifier is learned based on human-demonstrations (Shekhar et al., 2019), and systems that decide on the for questions based on its own uncertainty (Abbasnejad et al., 2019; Chi et al., 2020).

Previous works argued that a model that generates information-seeking questions should not be decoupled from the model that makes the final prediction for the task (Testoni and Bernardi, 2021; Abbasnejad et al., 2019). The model uncertainty over class labels can be used to decide on the content of the clarification question.

We use the same reasoning in our work, since the acting agent's uncertainty, used to decide *when* to ask a question, is more interpretable than a black-box model that is trained to make a positive prediction if a human decided to ask a question.

Our setup involves a human architect agent and an artificial builder agent situated in a 3D Grid-world environment. The human architect agent has access to a target structure and can direct the artificial builder agent with natural language.

A natural way of evaluating if an instruction-

following agent is asking questions at the right time is to use human evaluation, which is labor-intensive. In this work, we perform evaluation using already existing datasets (Narayan-Chen et al., 2019; Kiseleva et al., 2022a). We investigate if the builder model's uncertainty is higher for instructions that precede questions in these human-human dialogues.

Our contributions are:

1. We identify the uncertainty metrics that achieve the largest distributional separation between ambiguous and non-ambiguous instructions.

2. We show that calibration helps increase the statistical separation between the uncertainty metrics of ambiguous and non-ambiguous instructions.

3. We provide a novel empirical analysis on the relationship between uncertainty and dialog history length. We show uncertainty increases as length increases, confounding the statistical separation that is important for detecting the need for questions. This poses an inherent difficulty as in collaborative dialogues, the history grows over the course of interaction.

## 2 Methodology

We begin by defining ambiguous instructions and propose casting them as out-of-domain input in our modelling effort. We then describe our probabilistic model and the uncertainty metrics used for detecting the need for questions.

In this study, we use unsupervised methods for uncertainty estimation: We do not use examples of human questions in order to create a model for uncertainty. Instead, we obtain our measures from statistics that are obtained from the predictive probability distribution of our model.

### 2.1 What are ambiguous instructions?

Ambiguous sentences allow for multiple distinct interpretations. Depending on the domain, previous research has further refined this notion by contrasting "under-specification" with "ambiguity" (Clarke et al., 2009) or "ambiguity" with "inaccuracy" (Zukerman et al., 2015).

Our definition of ambiguity is based on utterances that prompted instruction-level clarification questions during a collaborative building game between humans in a grid-world environment. A dominant source of communication errors in this task is under-specification, e.g., the architect does not specify the color 3, or number of blocks. Other errors stem from mistakes in referring expressions that apply to multiple structures on grid, or none of them, e.g., using the wrong color. We subsume all of these cases that lead to clarification requests under "ambiguity".

**Ambiguous instructions are out-of domain data.** In our model, we cast ambiguous instructions as out-of-domain data and treat clear instructions as in-domain. To achieve this, our agent learns from demonstrations where the linguistic input was deemed actionable by humans. At test time, we present the agent with both clear and ambiguous utterances and measure its predictive uncertainty over the action space (Arora et al., 2021). Section 4 discusses a simple single-word messaging-game to illustrate how ambiguous messages can be viewed as out-of-domain examples in a grounded communicative setting.

### 2.2 Instruction following as prediction

In our first experimental setting (Section 4), we model our agent using a binary classification model. We use negative log-likelihood (NLL) and the probability of the predicted label (one-best probability) as uncertainty metrics.

In our second experimental setting (Section 5), we model our agent using a structured prediction (autoregressive) model. Using beam-decoding, the builder agent produces a sequence of actions that involves placing and removing colored blocks.

After each sequence prediction, we have access to multiple statistics from the beam: the likelihood of all the hypothesis in the beam, the likelihood of each token and the probability distribution over the whole vocabulary for each token in the beam. Previous work has investigated a number of ways to compute uncertainty measures from these beam-statistics (Malinin and Gales, 2021).

We formulate an ensemble-based uncertainty estimation framework which produces ensemble joint probabilities that tend to be well-calibrated (Lakshminarayanan et al., 2017). We consider an ensemble of $M$ models $\left\{\Pr\left(\mathbf{y} \mid \mathbf{x}; \boldsymbol{\theta}^{(m)}\right)\right\}_{m=1}^{M}$ where each model $\boldsymbol{\theta}^{(m)}$ captures the mapping between variable-length instruction $\mathbf{x} \in \mathcal{X}$ and sequence of actions $\mathbf{y} \in \mathcal{Y}$. The ensemble joint probabilities are obtained via a token-level ensemble:

$$\Pr(\mathbf{y} \mid \mathbf{x}; \boldsymbol{\theta}) = \prod_{l=1}^{L} \frac{1}{M} \sum_{m=1}^{M} \Pr\left(\mathbf{y}_l \mid \mathbf{y}_{<l}, \mathbf{x}; \boldsymbol{\theta}^{(m)}\right), \quad (1)$$

where $L$ is the length of the output sequence and $\boldsymbol{\theta} = (\boldsymbol{\theta}^{(1)}, \dots, \boldsymbol{\theta}^{(m)})$. Our agent learns from demonstrations where linguistic input was acted on by humans. At test time, we evaluate the model on ambiguous instructions that are not present during training. We quantify predictive uncertainty over the action space $\mathcal{Y}$ and investigate the distributional separation between the predictive uncertainty of ambiguous and non-ambiguous instructions. We use beam-search in the inference stage to provide high-quality hypotheses and compute uncertainty metrics based on the top hypotheses. We indicate the number of hypotheses used by the metrics below by the designation **N-best**, where $N \in \{1, 2, 5\}$. We next present the uncertainty measures used in our second experiment.

**Likelihood-based metrics (LL).** Negative log-likelihood (NLL), $-\log \Pr(\mathbf{y} \mid \mathbf{x}; \boldsymbol{\theta})$, is a proper scoring rule (Gneiting and Raftery, 2007), where an optimal score corresponds to perfect prediction. We compute two metrics based on NLL:

1. NLL of the top hypothesis (**1-best**).

2. NLL difference between the top two hypotheses (**2-best**). This has been used to guide question-asking for agents in Chi et al. (2020).

**Entropy-based metrics (ENT).** Following Malinin and Gales (2021), we compute a sequence-level Monte-Carlo approximations of entropy (**5-best**):

$$\mathcal{H}_S(\Pr(\mathbf{y} \mid \mathbf{x}; \boldsymbol{\theta})) \approx - \sum_{b=1}^{B} \frac{\pi_b}{L^{(b)}} \log \Pr(\mathbf{y}^{(b)} \mid \mathbf{x}; \boldsymbol{\theta}), \quad (2)$$

where the log conditional is scaled by:

$$\pi_b = \frac{\exp \frac{1}{T} \log \Pr(\mathbf{y}^{(b)} \mid \mathbf{x}; \boldsymbol{\theta})}{\sum_{k=1}^{B} \exp \frac{1}{T} \log \Pr(\mathbf{y}^{(k)} \mid \mathbf{x}; \boldsymbol{\theta})}. \quad (3)$$

The hyperparameter $T$ is a calibration temperature that is useful to adjust the contribution of lower-probability hypotheses to the entropy estimate. We also compute a token-level approximation (**5-best**):

$$\mathcal{H}_T(\Pr(\mathbf{y} \mid \mathbf{x}; \boldsymbol{\theta})) \approx \quad (4)$$

$$- \sum_{b=1}^{B} \sum_{l=1}^{L^{(b)}} \frac{\pi_b}{L^{(b)}} \mathcal{H}\left(\Pr(\mathbf{y}_l \mid \mathbf{y}_{<l}, \mathbf{x}; \boldsymbol{\theta})\right). \quad (5)$$

**KL-divergence-based metrics.** Following Lakshminarayanan et al. (2017), we compute the following "disagreement" metric (**1-best**):

$$\frac{1}{M} \sum_{m=1}^{M} \text{KL}\left(\Pr(\mathbf{y} \mid \mathbf{x}; \boldsymbol{\theta}^{(m)}) \,\|\, \Pr(\mathbf{y} \mid \mathbf{x}; \boldsymbol{\theta})\right). \quad (6)$$

We compute **LL** and **ENT** metrics for the ensemble model $\Pr(\mathbf{y} \mid \mathbf{x}; \boldsymbol{\theta})$ and the most accurate member of the ensemble.

## 3  Related Work

The need for clarification has received a lot of interest in the field of conversational search and dialog-systems. In spoken dialog-system, the confidence of the speech-recognition system is a popular starting point for prompting clarifications (Ayan et al., 2013; Bohus and Rudnicky, 2005). The ASR-confidence is often further refined by input from downstream processing such as the output of a natural language understanding module (Stoyanchev and Johnston, 2015; Komatani and Kawahara, 2000) or structured discourse-modeling (Schlangen, 2004). These more structured semantic representations often allow for custom notions of ambiguity, such as too many possible referents (Zukerman et al., 2015), or contradictions in the discourse representation (Skantze, 2008).

In recent years, we have witnessed a rising popularity of neural sequence-to-sequence implementations of conversational (OpenAI, 2023) and situated agents (Padmakumar et al., 2022; Wang et al., 2023). Given their end-to-end nature, more approaches model the need for clarification in these systems as a separate neural model, that is trained on data labelled with ambiguity tags (Kim et al., 2021; Aliannejadi et al., 2019, 2021). Closest to our domain, the model for timing clarification questions introduced by (Shi et al., 2022) also consists of a neural binary classifier trained using the occurrences of human clarification questions in dialog. They use the same 3D grid world and human data collection that we utilize in this work.

## 4  Ambiguous Messages in a Lewis' Signaling Game

We start our analysis with a simple speaker-listener game. We use this setup to illustrate the relation between calibration and detection of ambiguous instructions. In this referential task, the speaker and

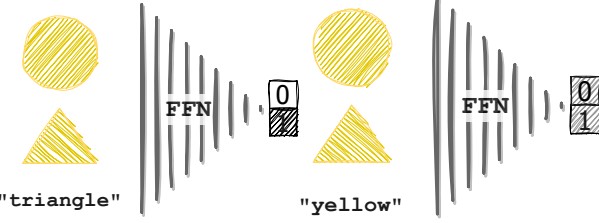

(a) Clear message "triangle"

(b) Ambiguous message "yellow"

Figure 1: Examples of a clear and an ambiguous message in the referential game. The two possible referents are a yellow circle and a yellow triangle

|  | clear | | ambiguous | |
|---|---|---|---|---|
|  | NLL | Accuracy | NLL | Accuracy |
| BASE | 0.005 | 1.0 | 1.71 | 0.5 |
| ENS | 0.006 | 1.0 | 0.71 | 0.5 |

Table 1: Average negative log-likelihood (NLL) and accuracy of the baseline model (BASE) and the ensemble model (ENS) evaluated on the in-domain clear instructions and the out-of-domain ambiguous instructions

the listener observe an ordered set of objects. The speaker chooses a message to refer to one of these objects and the listener has to correctly identify the referential target based on the message.

In our setup, only the listener agent is learned, and we restrict the task to a choice between two objects: $O_1$ and $O_2$. Each object is described by $c = 2$ properties and each property is a one-hot vector of $f = 4$ different values. In our visualization, the properties correspond to color and shape. The vocabulary is the set of all $v = c \times f = 8$ properties. During training, the speaker sends a single word that identifies the target object, and the listener is rewarded if it picks the correct one. See Figure 1 for an illustration of the setting.

The listening agent estimates the following binary probability distribution of identification $I$:

$$P(I|\langle O_1, O_2 \rangle, M), \qquad (7)$$

where message $M$ is a one-hot vector of size $v$ and each object is described by the concatenation of two one-hot vectors $O = \langle C, S \rangle$, where $C$ and $S$ are both of size $f$.

**Implementation details of the listener** In this simple simulated data experiment, we uniformly sample shapes, colors, and messages, rejecting cases where the message is ambiguous. The dataset has 960 unique unambiguous examples and we reserve 50 of these for evaluation.

The listener is parameterized with a feed-forward neural network that makes a binary decision over the indices of the two potential target objects. We train ten models with different random seeds in order to create a deep-ensemble model. As baseline (BASE), we randomly pick one of the ensemble member models since all models achieve perfect accuracy on an unseen combination of properties and instructions.

**Evaluating ambiguous instructions** We create ambiguous instructions for the listener by choosing a message describing a property that is shared by both possible referents. See Figure 1 for an example of a clear (a) and an example of an ambiguous (b) message. The desired model behaviour for ambiguous instructions is to have maximum predictive uncertainty. In this case, this means that the model should predict a probability of $0.5$ for both possible outcomes.

We evaluate the baseline (BASE) and the ensemble model (ENS) on both the clear in-domain and the ambiguous examples. The ambiguous instructions should be considered out-of-domain, as no such examples were presented during training. As shown in Table 1, both models have the same accuracy under both conditions, but the ensemble model (ENS) has a lower negative log-likelihood than the baseline (BASE) on the ambiguous examples, indicating an improvement in uncertainty quantification. In Figure 2, we observe that the ensemble model achieves a much better statistical separation (blue vs orange) between the one-best predicted probabilities of the in- and out-domain-data. The ensemble model's predicted probabilities also lie much closer to the expected value of $0.5$.

This indicates that the well-calibrated uncertainty estimates of the ensemble model increases the efficacy of using uncertainty metrics to detect ambiguity in the message.

## 5 Ambiguous Instructions in Grounded Collaborative Dialog

For our grounded dialog setting, we focus on agents trained to build structures in 3D environments based natural language instructions from a human partner. The corresponding data was collected during a series of collaborative building tasks where a human architect was giving directions to a human builder in order to build a target structure. During training, we present the builder agent with: (a) the relevant dialog history, including utterances from

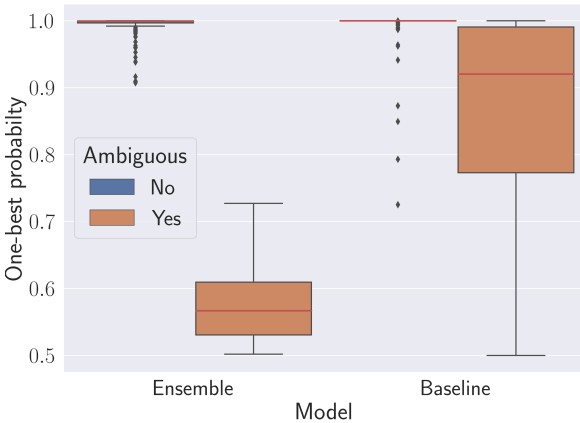

Figure 2: One-best probabilities of the predicted class for the ambiguous and non-ambiguous cases for each model (ensemble vs single model baseline).

both the architect and the builder, and (b) the state of the grid built in previous steps. The builder is trained to place and remove colored blocks using supervised signals from action sequences recorded from human-human interactions.

## 5.1 Datasets

We conduct our experiments on three existing datasets, and create a novel evaluation set that allows for a fine-grained analysis of ambiguity detection performance.

The Minecraft Dialogue Corpus (**MDC**) (Narayan-Chen et al., 2019) is a collection of 509 game-plays between two humans. This dataset was released with standard splits. We use the train portion of the data to train our builder agent and the test portion for evaluation. The test-set contains 1,827 clear dialog-turns and 177 questions.

The 2022 IGLU NLP task data collection (**IGLU-NLP**) (Mohanty et al., 2022) contains single-turn interactions between two humans. The architect is presented with an already built grid state and can add a single instruction to modify the grid which is then implemented by the builder. As we are interested in dialog-phenomena, we use this collection only for training.

**IGLU-MULTI** (Kiseleva et al., 2022a) is also a multi-turn data-collection similar to **MDC**. As it is a smaller dataset with a total of 667 dialog- and building turns across 120 game-plays, we reserve it for evaluation. The dataset contains 584 dialog turns for building and 83 clarification questions.

**Creating ambiguous instructions**

Both **IGLU-MULTI** and **MDC** (Shi et al., 2022) have been annotated in previous work with labels

for instruction-level clarification questions. While these annotations are helpful for evaluating overall ambiguity detection performance, they preclude a fine-grained analysis of agent uncertainty.

To this end, we create a new artificial dataset (**IGLU-MINI**) by manipulating the clear portion of the **IGLU-MULTI** dataset. We construct ambiguous examples using three main categories that we identified as reasons for questions in the original dataset: The color is not specified, the number of blocks is not specified, or there is an error when referencing the already built blocks.

For the first two categories, we manipulate the instruction by removing the color or the number of blocks. The third type of ambiguity is created by corrupting the intended referent of referring expressions by changing the color of the referenced blocks. We identify instructions that contain a reference to the already built grid by matching strings containing referring expressions starting with a definite article followed by a color in the first sentence of the instruction ("the red ones" in Figure 4).

We present examples of the resulting minimal pairs with their corresponding model output in Figure 4. This way we can generate pairs of clear and ambiguous instructions: We start with a clear instruction then perform the minimal edit to make it defective. We manually checked the generated examples to make sure that they remain grammatical.

## 5.2 Implementation

The builder's task is formulated as a sequence-to-sequence prediction task. This is achieved by converting the grid-state into a textual description. The grid is described by the sequence of actions that are necessary to build it. For example, building two green blocks diagonally corresponds to the string *"put initial green block. put green 1 left and 2 before."*. Actions are separated by a dot. All locations are expressed by their relative distance to the first building block. There are 6 tokens corresponding to moving in the negative or positive direction on $x$ (*left*, *right*), $y$ (*lower*, *upper*) and $z$ (*before*, *after*) axes. This parameterization of grid traversal is based on the approach of in Kiseleva et al. (2022b) [1].

We train a 10-member ensemble model using the T5-small (Raffel et al., 2020) transformer architec-

---

[1] See https://gitlab.aicrowd.com/aicrowd/challenges/iglu-challenge-2022/iglu-2022-rl-mhb-baseline for the starter code and the GridWorld environment (Zholus et al., 2022).

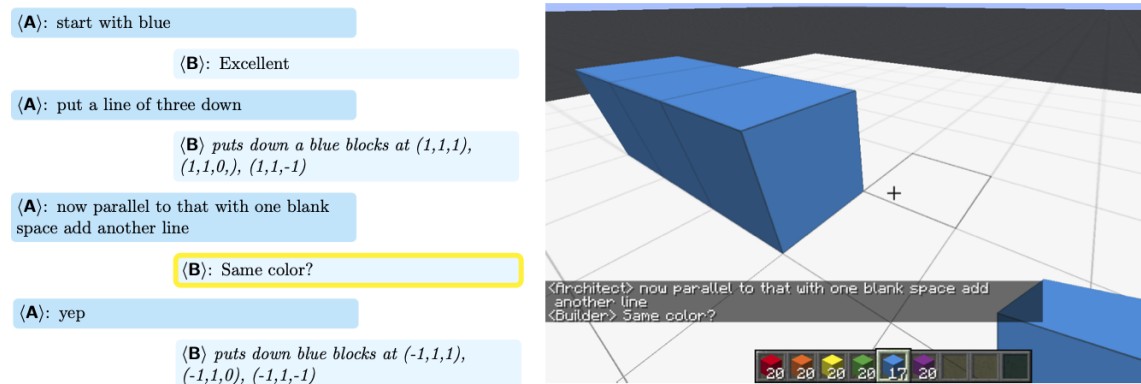

Figure 3: Example interaction from the IGLU-MULTI dataset with clarification question.

| | F1 | | NLL | |
|---|---|---|---|---|
| | ENS | Single | ENS | Single |
| IGLU-MULTI | 0.34 | 0.35 | 0.35 | 0.4 |
| MDC | 0.36 | 0.34 | 0.30 | 0.359 |

Table 2: F1 defined using block accuracy and recall of the ensemble (ENS) and its strongest member (Single). While performance is comparable for both models, ENS achieves lower negative log-likelihood on both datasets.

ture. Five members of the ensemble are fine-tuned from a pretrained T5-small with different random seeds. In order to create higher diversity within the ensemble, we train five further T5-small models from scratch with different random initialization. Fine-tuned models achieve better accuracy, while randomly initialized models introduce greater variance within the ensemble, which in turn leads to better calibration on the evaluation set. All models are trained for ten epochs, and we select the checkpoints with the best performance on the validation portion of the **MDC** dataset. We pick the best performing single and ensemble model respectively for further analysis [2].

### 5.3 Experiments

#### 5.3.1 Performance on clear instructions

Table 2 shows the performance of the ensemble and the baseline model. The performance on the block accuracy and recall matches F1 scores reported in previous work (Shekhar et al., 2019; Kiseleva et al., 2022b). We observe a similar picture as in the messaging game in Section 4:

#### 5.3.2 Detecting ambiguity at a dataset level

To compare the builder's predictive uncertainty in the clear and ambiguous conditions, we evaluate

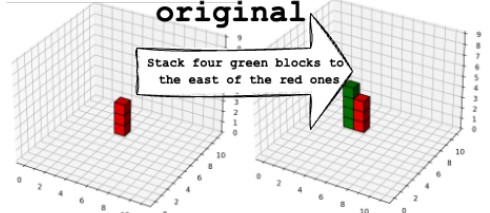

(a) Original example from the dataset.

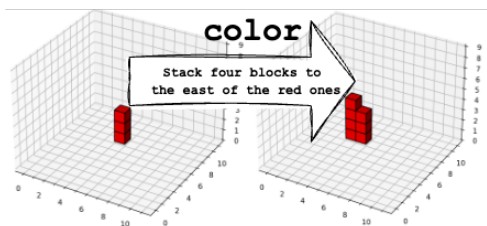

(b) The color of the blocks is removed from the instruction.

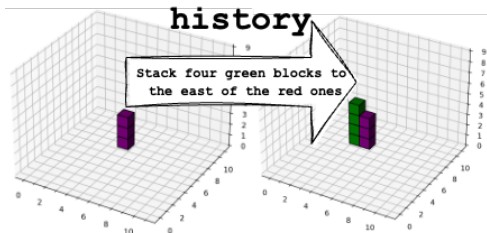

(c) The referenced structure is recolored from red to purple.

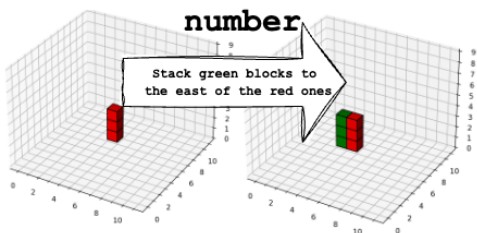

(d) The size of the column is removed from the instruction.

Figure 4: Three ambiguous instructions created with minimal edits. The starting grid is presented on the left and the corresponding model output is on the right.

---

[2]The implementation for the builder agents in this paper is available at https://github.com/naszka/uncertain_builder.

|   | Measure | Level | Aggr. | N-best | IGLU ENS | IGLU Single | MDC ENS | MDC Single |
|---|---------|-------|-------|--------|----------|-------------|---------|------------|
| 0 | ENT | SEQ | - | 5-best | 0.57** | 0.54** | 0.54 | 0.49 |
| 1 | ENT | TOK | avg | 5-best | 0.56* | 0.50 | 0.65** | 0.59** |
| 2 | KL | TOK | avg | 5-best | 0.63** | - | 0.59** | - |
| 3 | LL | SEQ | - | 1-best | 0.56** | 0.53 | 0.62** | 0.55* |
| 4 | LL | SEQ | diff | 2-best | 0.53 | 0.51 | 0.59 | 0.57 |

Table 3: Area under the ROC curve on the **IGLU-MULTI** and **MDC** datasets.

the model on all builder-turns: when the builder decided to build a sequence of blocks in the original dataset, and also when the builder asked a question. We compute the area under the receiver operating characteristic curve to show how well each uncertainty measure separates the clear from the ambiguous architect instructions. The significance thresholds (* for $p < 0.05$, ** for $p < 0.01$) were computed with a permutation test. The null hypothesis associated with this test is that the observed uncertainties come from the same distribution for the ambiguous and non-ambiguous examples. Table 3 shows that while the ensemble model achieves slightly better separation across all uncertainty measures, both models struggle to separate cases where the human participants asked a question and where they decided to carry out the instruction.

### 5.3.3 Inducing ambiguity via minimal edits

We would like to gain a better understanding what drives the changes in predictive uncertainty of our builder. The **IGLU-MINI** dataset offers a controlled environment to investigate this question. Here we have pairs of instructions: the original one can be found in the human-human recordings and the builder deemed it clear enough to act upon it. The corresponding instruction has been minimally edited to increase linguistic confusion. In Figure 4 we illustrate each group of minimal changes we introduced. In this setup, we can ask a more targeted question: Does higher linguistic ambiguity translate to higher uncertainty in the builder? Instead of comparing all clear instruction with all ambiguous ones, here we can examine the uncertainty changes within each pair. In this setting, both the number of prior dialog turns and the size of the already built structure are kept constant.

We expect each type of minimal-edit to translate differently in the predictive uncertainties. We introduce some new features to better capture these effects. Where we omit the **color** for the original instruction, we expect the change of uncertainty to concentrate on the predicted color tokens. Hence,

|   | Position | Measure | Level | Aggr. | N-best | First act. | ENS | Single |
|---|----------|---------|-------|-------|--------|------------|-----|--------|
| 0 | all | ENT | TOK | avg | 5-best | No | 0.92** | 0.83** |
| 1 | all | ENT | SEQ | - | 5-best | No | 0.92** | 0.80** |
| 2 | all | LL | SEQ | - | 1-best | No | 0.89** | 0.79** |
| 3 | color | ENT | TOK | max | 5-best | No | 0.95** | 0.90** |
| 4 | color | ENT | TOK | avg | 5-best | No | 0.94** | 0.92** |
| 5 | color | LL | TOK | min | 5-best | No | 0.94** | 0.89** |

(a) **Color**-ambiguity pairs.

|   | Position | Measure | Level | Aggr. | N-best | First act. | ENS | Single |
|---|----------|---------|-------|-------|--------|------------|-----|--------|
| 0 | all | ENT | TOK | avg | 5-best | No | 0.79** | 0.62** |
| 1 | all | ENT | SEQ | - | 5-best | No | 0.79** | 0.53 |
| 2 | all | LL | SEQ | - | 1-best | No | 0.78** | 0.60** |
| 3 | eos | ENT | TOK | - | 1-best | Yes | 0.88** | 0.63** |
| 4 | eos | ENT | TOK | avg | 5-best | No | 0.84** | 0.63** |
| 5 | eos | LL | TOK | avg | 5-best | No | 0.84** | 0.64** |

(b) **Number**-ambiguity pairs.

|   | Position | Measure | Level | Aggr. | N-best | First act. | ENS | Single |
|---|----------|---------|-------|-------|--------|------------|-----|--------|
| 0 | all | ENT | TOK | avg | 5-best | No | 0.66* | 0.70** |
| 1 | all | ENT | SEQ | - | 5-best | No | 0.72* | 0.72** |
| 2 | all | LL | SEQ | - | 1-best | No | 0.66* | 0.67** |
| 3 | color | ENT | TOK | avg | 5-best | No | 0.83** | 0.56 |
| 4 | all | ENT | TOK | avg | 5-best | Yes | 0.72** | 0.65* |
| 5 | color | LL | TOK | avg | 5-best | No | 0.72* | 0.53 |

(c) **History**-ambiguity pairs.

Table 4: Proportion of examples in the **IGLU-MINI** dataset where the ambiguous instruction has higher uncertainty than the corresponding pair according to the more calibrated ENS and the baseline Single models. Rows 0-2 show the most commonly used structured prediction metrics. In rows 3-5, the three best performing metrics are displayed. The ensemble model outperforms the uncalibrated single model in detecting the hike in semantic difficulty.

we compute different statistics only considering these positions of the sequence.

When we leave out the **number** of blocks to be used, we expect the model to struggle with deciding if it should start a new action after the end-of-action symbol. To detect this, we focus the measurements on positions where there could be an end-of-sequence token, namely positions right after each end-of-action token.

We also compute all our measurements limited to only the first action of the predicted action-sequence. We expect that where the main reference point on the grid has been altered (**history**), most of the uncertainty change will be observed on where to place the first block.

In Table 4, we compare how many times the utterance with increased semantic difficulty has resulted in higher uncertainty. The table also shows the result for paired permutation test, which indicates if the change in the uncertainty measurements is significant.

For each category of semantic error, different measures of uncertainty perform best. For color-ambiguity, the top performing measures only consider the color-predictions. In case of number-obfuscation, all best performing metrics consider

positions where the end of sentence (EOS) could be predicted (after each action). The best performing measure is the entropy of the token-level distribution after the very first action in each 5-best hypothesis. This shows that the models struggle to decide if a second action should be started. When we alter the color of the referenced structure, the best performing metric considers the entropy of all the color-predictions. This is explained by the model's reliance on the color of built structures to decide on the color.

## 6 Analysis

While for ambiguity minimal-pairs, the ensemble model can confidently differentiate between the corresponding instructions in terms of uncertainty, the within-class variance remains high, rendering it very difficult to find an overall threshold that separates clear- from ambiguous cases (Fig. 5).

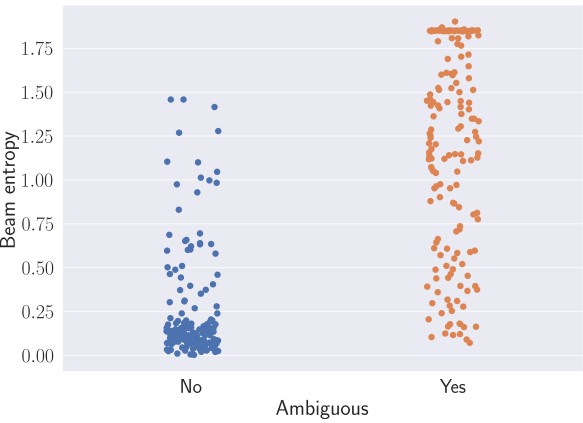

Figure 5: Uncertainty of the original (blue) and the corresponding ambiguous instruction (orange) for the ensemble model. While uncertainty is higher than the corresponding baseline utterance within pairs, within-class variance is very high.

The main driver of predictive uncertainty is the length of the input sequence. Figure 7 illustrates the correlation between input length and uncertainty for both models. The input length consists of the token length of the dialog history and the size of the already built structures in the grid. As a result, the uncertainty of the builder steadily increases over the dialog, even if each step was understood well by the builder and no questions were asked.

For significantly longer input sequences, the performance of the builder model decreases, for these cases, it is expected that uncertainty of the model rises. In Figure 6, we focus our attention on input sequences under the length of 250. While the performance is close to constant in this range, the

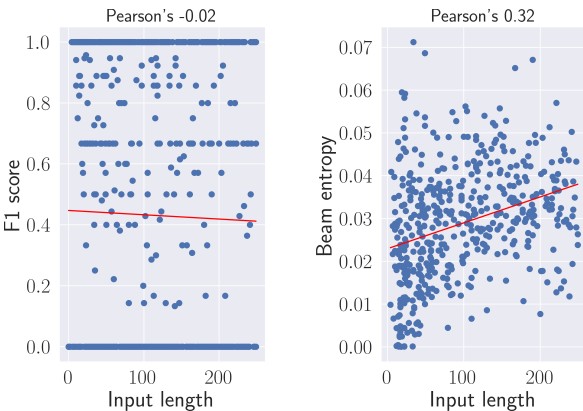

Figure 6: Correlation of the input length with the F1 score (left) and the entropy of the prediction (right) for all sequences under a length of 250. While the performance is constant, the entropy of the predictions increases with the input length.

entropy still increases with the input length.

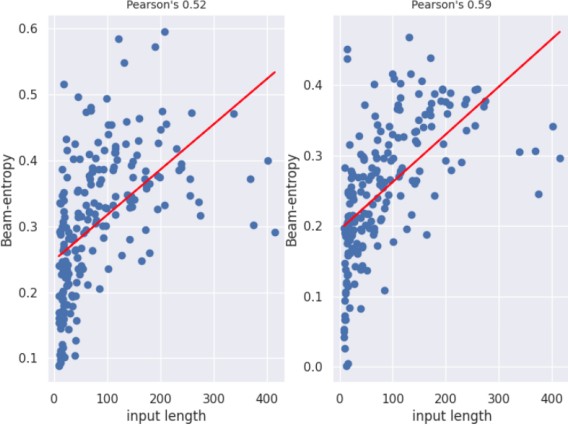

Figure 7: Correlation of input length with the prediction entropy in ensemble (left) and the baseline model (right). The input length consists of the token length of the dialog history and the size of the already built structures.

## 7 Conclusion and Future Work

In this work, we have investigated the alignment between predictive uncertainty and ambiguous instructions in visually grounded communication. We show that calibration can help increase the efficacy of unsupervised uncertainty metrics. However, our empirical analysis on the relationship between uncertainty and dialog history length also shows that varying input lengths can pose a problem for using uncertainty metrics for detection. At the same time, these metrics do exhibit the expected rise in uncertainty when applied to detecting targeted ambiguity changes.

The task of detecting the need for questions in this environment has been posed in the frame of the

2022 IGLU NLP challenge (Kiseleva et al., 2022b). Crucially, the evaluations of the builder agent and question asking module have been separated into two tracks. For future work, we can use the uncertainty metrics generated by the builder to detect the need for asking questions, removing the need for a bespoke model for the latter.

Highly varying input-lengths is an inherent feature of dialogue. The uncertainty measures that we utilised in this paper, have been previously shown to be effective in detecting out-of-domain samples in POS-tagging (Ulmer et al., 2022), ASR (Tu et al., 2022) and NMT (Malinin and Gales, 2021). These applications do not exhibit the same magnitude of variance in the length of the model's input.

## 8 Limitations

We examined one specific way of presenting the IGLU builder task data to a sequence-to-sequence transformer, introduced in prior work. There are potentially many alternative methods for casting this data as a sequence-to-sequence problem, and our observations might not hold for other data-transformations or architectures.

The data-collections for our considered tasks are very small compared to the scale of data used for state-of-the art conversational systems (OpenAI, 2023). It has been pointed out in related work that the calibration properties of large DNNs improve greatly with higher in-domain accuracy and more training data (Desai and Durrett, 2020).

## 9 Acknowledgements

We thank Taewoon Kim and Michiel van der Meer for early brainstorming on this project. We are also grateful to Alberto Testoni, Mohamed Omran and the anonymous reviewers for their valuable feedback. This research was funded in part by the Netherlands Organization for Scientific Research (NWO) under project number VI.C.192.080. We also received funding from the Hybrid Intelligence Center, a 10-year programme funded by the Dutch Ministry of Education, Culture and Science through the Netherlands Organisation for Scientific Research with grant number 024.004.022.

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
