# OpenReview forum: "Aligning Predictive Uncertainty with Clarification Questions in Grounded Dialog"
_EMNLP/2023/Conference — EMNLP 2023 Findings_

### Official Review · Reviewer_tDZx · 2023-08-01

**Typos Grammar Style And Presentation Improvements:** 466 ... [of] what drives ...
**Soundness:** 4

**Excitement:**

4: Strong: This paper deepens the understanding of some phenomenon or lowers the barriers to an existing research direction.

**Paper Topic And Main Contributions:**

This paper addresses the problem of when to ask clarifying questions in grounded dialog, i.e. due to some ambiguity in the user's instruction / goal. The paper introduces several unsupervised methods that can be used to compute the model's uncertainty, and performs a series of experiments to see how well this is aligned with genuinely ambiguous instructions. Empirically, they demonstrate this first with a toy example via Lewis' signalling game, then with the Minecraft Collaborative Building Task. Overall, they show the metrics are generally well-correlated with uncertainty, although they also identify input length as a confounding factor which seems to cause uncertainty to increase.

**Questions For The Authors:**

(1) For the ensemble model, how do you ensure the fine-tuned T5 models are different--is this just by changing the random seed?

Post-rebuttal edit:

I thank the authors for answering the question around implementation of the ensemble.

**Reasons To Accept:**

(1) The proposed method has greater interpretability than using a separate classifier to predict uncertainty, as the uncertainty is derived from the generative model's own probability distribution.

(2) The proposed method is unsupervised, so does not require labelled examples of uncertainty.

(3) The proposed method leverages properties that can be readily computed on most off-the-shelf generative transformers (e.g. NLL), so could easily be applied to many existing use cases.

(4) There is extensive empirical detail provided to support the paper's claims.

**Reasons To Reject:**

(1) I would have liked to see more detail on how an ensemble is implemented with the T5 pretrained model, i.e. given each model starts from the same set of weights, how do we ensure the finetuned models are different?

(2) There isn't much reference made to existing techniques on detecting out-of-domain inputs outside of the introduction, so it is hard to compare how novel / effective the proposed metrics are.

(3) The tasks seem somewhat artificial due to the clear boundary between ambiguous versus unambiguous instructions. Most real life instructions are not entirely unambiguous and rely on a system selecting the most plausible hypothesis, e.g. "Book the restaurant tomorrow at 6" could mean 6pm or 6am, but contextually is probably 6pm. In these types of cases, it is unclear how robust the proposed metrics would be.

Post-rebuttal edit:

Issue (1) was resolved by the author response (random seeds changing order that training data is shown).  For issue (2) the authors say this is due to limited work addressing the same problem as the authors--unfortunately I am not well enough versed in the specific literature to verify this, so I leave this point to other reviewers. Issue (3) is perhaps the largest issue, as other reviewers have raised similar points--namely, that the scenario being presented may be too artificial and therefore it is unclear how well the techniques may transfer on to the original task.

I have left my rating at 4/4, although I agree with the concerns from other reviewers about the setting being artificial and that the results in even in that setting are not decisive. The fact that there is no clear line between ambiguous and non-ambiguous in Figure 6, even though the dataset is artificial suggests that the method would struggle even more in real-life datasets, where the boundary between ambiguous and non-ambiguous is less clear cut. It may be possible to introduce various normalisation techniques to obtain better separation, such as accounting for dialogue history length, which empirically seems to be a confounding factor; however, this may undermine the point of interpretability, which is one of the paper's current strengths.

**Reproducibility:**

3: Could reproduce the results with some difficulty. The settings of parameters are underspecified or subjectively determined; the training/evaluation data are not widely available.

**Reviewer Confidence:**

3: Pretty sure, but there's a chance I missed something. Although I have a good feel for this area in general, I did not carefully check the paper's details, e.g., the math, experimental design, or novelty.

---

> ### Author Rebuttal · Authors · 2023-08-29
>
> Thank you for your valuable feedback tDZx. We are happy to hear that you appreciate the **advantages of unsupervised techniques in semantic ambiguity detection** and **find our empirical analysis extensive**. We address your remarks and questions below.
>
> **Remark 1 and Question 1:**
>
> > I would have liked to see more detail on how an ensemble is implemented with the T5 pretrained model,
>
> > (1) For the ensemble model, how do you ensure the fine-tuned T5 models are different--is this just by changing the random seed?
>
> We fine tune the T5 model five times with different random seeds. The choice of seed determines an ordering of the training data. In order to increase diversity in the ensemble, we also trained five models from scratch. We use the AdamW optimizer with a starting learning rate of $1e-5$ . All models are trained for 10 epochs and we use the checkpoints with the best performance on the validation set. For validation, we use a concatenation of the IGLU-MULTI dataset  and the validation portion of the MDC dataset. As the table below indicates, this is crucial for better calibration.
>
> We will add more details about the ensemble creation in the final manuscript. The code with all the random seeds and model checkpoints necessary to exactly reproduce the results will be released.
>
> |                                        | IGLU-MULTI avg NLL |
> |----------------------------------------|------------------------|
> | 1 fine-tuned model                     | 0.41                   |
> | 3 fine-tuned models                    | 0.367                  |
> | 5 fine-tuned models                    | 0.354                  |
> | 5 fine-tuned and 5 from scratch models | 0.231                  |
>
> **Remark 2:**
>
> > much reference made to existing techniques on detecting out-of-domain inputs outside of the introduction, so it is hard to compare how novel / effective the proposed metrics are.
>
> We are not aware of other datasets with a combination of properties of grounded dialog task that we consider in this paper: free-form, multi-turn natural language dialog containing questions, with grounding from another modality, and with a well-defined task goal. Hence, the probabilistic measures we investigate for the purpose of detecting ambiguity have not been previously evaluated in such a setting. However, they have been used for OOD-detection in other scenarios with varying results depending on the nature of distribution shift in the data ([1], [2], [3]). We currently have these works in the future work section, but we will make sure to discuss them in the related work of the manuscript.
>
>
> **Remark 3:**
>
> We agree with the reviewer that there can be varying degrees of ambiguity in each dialog turn. Some might be easier to resolve with further pragmatic reasoning than others. In the unaltered natural language dialogs (MDC and IGLU-MULTI) there might be an inherent variation between human preferences about what degree of ambiguity prompts them to ask questions. We do not currently address this problem in this paper and assume that on average, clarification questions indicate ambiguous inputs. This also applies for the artificial manipulations in the IGLU-MINI dataset. Some of the introduced ambiguities present more difficulty than others depending on the context. We briefly discuss, for example, that the model learns to use the most recent colour when the colour is not mentioned in the last dialog turn. We do not attempt to break down the dataset based on levels of ambiguity, but we assume that on average, we introduce higher levels of ambiguity. This is confirmed by the uncertainty estimates of the ensemble model.

---

### Official Review · Reviewer_pdGz · 2023-08-05

**Typos Grammar Style And Presentation Improvements:** Line 505
**Soundness:** 3

**Excitement:**

2: Mediocre: This paper makes marginal contributions (vs non-contemporaneous work), so I would rather not see it in the conference.

**Paper Topic And Main Contributions:**

This paper studies unsupervised metrics that can be used to measure the uncertainty of the requests. Based on preliminary analysis and experiments on human-machine collaboration tasks where the goal is to successfully follow the instruction of users and collaboratively build a target structure, the authors show that the proposed metrics can be used to identify when to ask clarification to the user. Also, the paper presents useful empirical findings that the better calibration leads to better detection and the length of the dialogue context is correlated with the uncertainty of model predictions.

**Questions For The Authors:**

* The "Lewis signaling game" part seems to be redundant the complexity of the task is too limited compared to the original tasks. Why should we analyze the behavior in this toy setting ? Do you have any special observation here?

* Do you have any experimental results varying the size of the training samples to see when do the uncertainty metrics become invalid?

* Do you have any results with different hyper-parameters like N ∈ {1, 2, 5}?


Post-rebuttal comments:
Thank you for providing the additional results and details. The questions are all resolved. I will update the reproducibility score.

**Reasons To Accept:**

* Detecting when to ask for clarification to the user is an important research topic since human-machine collaboration, especially with chatbots like ChatGPT is becoming very popular in the society.

**Reasons To Reject:**

* The contribution of the paper is very limited. The metrics are already very popular way of measuring uncertainty in other fields like active learning and the concrete method on how to use the metrics for classifying if a user request is vague is not suggested.

* The process on how to build the artificial dataset (IGLU-MINI) should be more clear and the example of the manipulation should be described in the paper since one of the main result (figure 5) can be biased when the examples in the dataset have some dataset artifacts.

* More details are required to reproduce the results, such as hyper-parameters used.

Post-rebuttal comments:
I've carefully read the authors' comments and I appreciate that some details are now clarified with the comments. However, there are critical concerns remaining still. The main empirical finding based on their artificial scenarios, namely the alignment between ambiguous linguistic inputs and the model’s predictive uncertainty, sounds less likely to be transferrable to real-world setup as other reviewers also mentioned. Also, in abstract of the paper, the authors said they "investigate if ambiguous linguistic instructions can be **detected** by uncertainty in neural models.", but I don't think we can make significant progress regarding the topic since the paper does not suggest concrete method to **detect** when the linguistic instructions are vague. We could just partially conclude that ambiguous linguistic instructions might be **related to** the uncertainty to some extent.

**Reproducibility:**

4: Could mostly reproduce the results, but there may be some variation because of sample variance or minor variations in their interpretation of the protocol or method.

**Reviewer Confidence:**

4: Quite sure. I tried to check the important points carefully. It's unlikely, though conceivable, that I missed something that should affect my ratings.

---

> ### Author Rebuttal · Authors · 2023-08-29
>
> Thank you for your valuable feedback pdGz.  We are happy to hear that you find our **empirical findings useful** and our **objective important for human-machine collaboration**. We address your remarks and questions below.
>
> > The metrics are already very popular
> > the concrete method on how to use the metrics for classifying if a user request is vague is not suggested.
>
>
> The main focus of this paper is not to propose a new method for detecting when to ask clarification questions, but to investigate the alignment between ambiguous linguistic inputs and the model’s predictive uncertainty.  The uncertainty metrics we use are well-studied and we seek to identify the ones that can potentially be useful for detecting when to ask clarification questions (Table 3) and whether calibration is important (Table 4).  To the best of our knowledge, in the structured data setting (such as in IGLU-MULTI and MDC), this has not been investigated.
>
> In this paper, we also show that the uncertainty increases with length (Figure 8), confounding the statistical separation that is important for detecting the need for questions.  This inherent difficulty is more apparent in collaborative dialogues, compared to related domains such as machine translation and automatic speech recognition.
>
> Lastly, our analysis provides principled yet simple baselines that give a probabilistic interpretation of linguistic ambiguity in a grounded environment using sequence-to-sequence models.
>
> We note that similar points have been raised by d5nS, so we will make it clear in our contributions list (Page 2, right before Section 2) that we do not seek to propose a new and concrete method, but to investigate the usefulness and behaviour of these uncertainty metrics for detecting clarification questions in the grounded, multi-turn dialog setting.
>
> > should be more clear and the example of the manipulation should be described in the paper
>
> Regarding the IGLU-MINI data, we create the minimal pairs by string manipulations. We remove the first word that is an exact match.  For example, for a given input `Stack three red blocks on top of each red block. Put one yellow block.` The first colour (red)  is removed when the colour entity is manipulated and the first number (three) is removed when the number entity is manipulated.
>
> For history manipulation, we first check if the instruction refers to the past by a regular expression that looks for definite articles in referring expressions in the first sentence of the last instruction. For example, “Place a green block on top of the red tower.”  refers to the built grid because “the red tower” is a referring expression with a definite article.
>
> After finding instructions that refer to the past, we replace all mentions of a colour with another colour in both the dialog history and the  built grid description. We will release the implementation of the data-generator for IGLU-MINI.
>
> As mentioned in the manuscript, we manually checked our examples to make sure they are grammatical. Lastly, both baseline (Single) and the ensemble (ENS) models use the same examples, so if the dataset manipulation creates artificial advantages for detecting ambiguity, both models would benefit from it.
>
> > More details are required to reproduce the results
>
> For the Signalling game, we use an RNN with 2 hidden layers and a hidden size of 256. This setup is not sensitive to the choice of hyper-parameters – the results are substantially the same across different learning rates, optimizers, and model sizes.
> For the IGLU building task, we use a pretrained T5-small and fine tune it five times with different random seeds. The choice of seed determines an ordering of the training data. In order to increase diversity in the ensemble, we also trained five models from scratch. We use the AdamW optimizer with a starting learning rate of 1e-5. All models are trained for 10 epochs and we use the checkpoints with the best performance on the validation set. For validation, we use a concatenation of the IGLU-MULTI dataset  and the validation portion of the MDC dataset.
> We will  include these details in the final manuscript. We will also release code for reproduction and the trained model checkpoints.
>
>
> # Answers to questions:
>
> >The "Lewis signaling game" part seems to be redundant the complexity of the task is too limited compared to the original tasks. Why should we analyze the behavior in this toy setting ? Do you have any special observation here?
>
> We use this example as a warm-up exercise about how OOD-detection and ambiguous messages relate to each other, this helps build a clear storyline as pointed out by reviewer 2.
> Further, the Lewis signalling game is  a popular setup for investigating basic patterns of grounded communication ([1, 2, 3]), but previous work never considered evaluation under ambiguous messages. It’s this paper's contribution to introduce this angle to the game.
>
> > Do you have any experimental results varying the size of the training samples to see when do the uncertainty metrics become invalid?
>
> Our analysis in Figure 7 shows that the influence of the input length already obfuscates the uncertainty measures for shorter inputs. We agree with the reviewer that investigating calibration properties of transformer models in different regions of input-length is an interesting and actionable next step based on our work.
>
> > Do you have any results with different hyper-parameters like N ∈ {1, 2, 5}?
>
> Yes. The Single baseline model corresponds to N=1. We also evaluated the average negative log-likelihood for fewer models on the IGLU-MULTI dataset, please see the table below. The final ensemble contains 5  fine-tuned T5-small models and 5 models trained from scratch with the same architecture. The models that are trained from scratch increase diversity ensemble and improves calibration.
>
> |                                        | IGLU-MULTI avg NLL |
> |----------------------------------------|------------------------|
> | 1 fine-tuned model                     | 0.41                   |
> | 3 fine-tuned models                    | 0.367                  |
> | 5 fine-tuned models                    | 0.354                  |
> | 5 fine-tuned and 5 from scratch models | 0.231                  |
>
>
>
> [1]  Julia White, Jesse Mu, Noah D. Goodman: Learning to refer informatively by amortizing pragmatic reasoning. CogSci 2020
>
> [2] Athul Paul Jacob, Mike Lewis, Jacob Andreas: Multitasking Inhibits Semantic Drift. NAACL-HLT 2021: 5351-5366
>
> [3] Ryan Lowe, Abhinav Gupta, Jakob N. Foerster, Douwe Kiela, Joelle Pineau: On the interaction between supervision and self-play in emergent communication. ICLR 2020

---

### Official Review · Reviewer_d5nS · 2023-08-11

**Soundness:** 3

**Excitement:**

2: Mediocre: This paper makes marginal contributions (vs non-contemporaneous work), so I would rather not see it in the conference.

**Paper Topic And Main Contributions:**

The paper studies whether neural model prediction uncertainty can be used to detect ambiguous instructions in grounded collaborative dialogs.

The authors first define a set of metrics quantifying prediction uncertainties, then they model the problem as a domain detection problem, with clear instructions in-domain data which model is trained on, ambiguous ones out-of-domain data.

In experiments, the authors demonstrate in a simple synthetic binary classification task, the prediction uncertainties from a FFN well correlates with whether the instruction is ambiguous, and the uncertainty from the FFN ensemble better detects the ambiguity.  Next the authors study two visually grounded dialog datasets, using whether the listener asks a clarification question as the label of ambiguous speaker instruction, but found the finetuned T5 model (and the ensemble) can't separate the two cases. Then the authors create a new synthetic dataset by making minimal edits of unambiguous instructions to create controllable ambiguous ones, and report that it's significant that the model's prediction uncertainty on an unambiguous instruction is lower than the corresponding ambiguous one, but one can still not use uncertainty to detect ambiguous instructions due to the high within class variance. Finally the authors argue that it's because the uncertainty is highly correlated with dialog history length.



**Reasons To Accept:**

The paper is clearly written and self-contained. The authors describe the methodology, a preliminary experiment, experiments with inconsistent results, and further in-depth study and analysis. It's very easy for readers to follow the clear storyline.

The problem the authors try to study, detecting ambiguous dialog turns, has good practical potentials in helping improve the quality of generated dialogs if solved.

**Reasons To Reject:**

- The paper is not able to give a solution to the problem it tries to solve. The proposed method works in the preliminary experiment, but there are still gaps for it to work in a synthetic dataset authors create with minimal variations, let along on the original task.

- The paper is using off-the-shelf uncertainty metrics and models, and the problem it trying to study is transformed into an OOD detection problem, but the paper doesn't bring any innovative ideas of solving the problem.

**Reproducibility:**

4: Could mostly reproduce the results, but there may be some variation because of sample variance or minor variations in their interpretation of the protocol or method.

**Reviewer Confidence:**

3: Pretty sure, but there's a chance I missed something. Although I have a good feel for this area in general, I did not carefully check the paper's details, e.g., the math, experimental design, or novelty.

---

> ### Author Rebuttal · Authors · 2023-08-29
>
> Thank you for your valuable feedback d5nS.  We are happy to hear that you find our work well-written, self-contained, and can potentially improve the quality of generated dialogs in practice.  We address your remarks below.
>
>
> The main focus of this paper is not to propose a new method for detecting when to ask clarification questions, but to investigate the alignment between ambiguous linguistic inputs and the model’s predictive uncertainty.  The uncertainty metrics we use are well-studied and we seek to identify the ones that can potentially be useful for detecting when to ask clarification questions (Table 3 and Table 5) and whether calibration is important (Table 4).  To the best of our knowledge, in the grounded, multi-turn dialog setting (such as in IGLU-MULTI and MDC), this has not been investigated.
>
>
> In this paper, we also show that the uncertainty increases with length (Figure 8), confounding the statistical separation that is important for detecting the need for questions.  This inherent difficulty is more apparent in collaborative dialogues, compared to related domains such as machine translation and automatic speech recognition. Additionally, our IGLU-MINI dataset allows analysis to be performed for each type of semantic ambiguity (Table 5a, 5b, and 5c) and we show that different measures of uncertainty attain the best performance.
>
> Lastly, our analysis provides principled yet simple baselines that give a probabilistic interpretation of linguistic ambiguity in a grounded environment using sequence-to-sequence models.
> We note that similar points have been raised by pdGz, so we will make it clear in our contributions list (Page 2, right before Section 2) that we do not seek to propose a new and concrete method, but to investigate the usefulness and behavior of these uncertainty metrics for detecting ambiguity  in the grounded, multi-turn dialog setting.

---

### Official Review · Reviewer_zqDm · 2023-08-12

**Soundness:** 4

**Excitement:**

4: Strong: This paper deepens the understanding of some phenomenon or lowers the barriers to an existing research direction.

**Paper Topic And Main Contributions:**

This paper focuses on the issue of dealing with uncertainty in visually grounded dialog between two agents, where underspecification in instructions can lead to ambiguity in following them and accomplishing a task. The authors alter examples from the IGLU-MULTI dataset to remove certain information from the instructions and study the effect this has on the receiving agent’s understanding. It presents metrics to quantify uncertainty to help identify ambiguous instructions, shows how ambiguous and unambiguous instructions can be statistically separated, and provides an analysis on how uncertainty increases with the duration of the dialog.

**Questions For The Authors:**

A. What was the rationale behind choosing an ensemble size of ten? Did you evaluate how the size of the ensemble impacts its performance?
B. What is architecture of neural network and hyper-parameters used to train it?
C. Did you consider embedding the instructions before and after the minimal edits to see how changes in their semantic similarity relate to uncertainty in the builder agent?

**Reasons To Accept:**

The paper presents a useful empirical analysis of the factors that lead to ambiguity in dialog instructions and uncertainty in following them, especially how this scales with the length of the dialog history. It is well-written and easy to follow.

**Reasons To Reject:**

No major reasons to reject.

**Reproducibility:**

3: Could reproduce the results with some difficulty. The settings of parameters are underspecified or subjectively determined; the training/evaluation data are not widely available.

**Reviewer Confidence:**

2: Willing to defend my evaluation, but it is fairly likely that I missed some details, didn't understand some central points, or can't be sure about the novelty of the work.

---

> ### Author Rebuttal · Authors · 2023-08-29
>
> Thank you for your valuable feedback zqDm.  We are happy to hear that you find our **empirical analysis useful**, **well-written** and **easy to follow**.  We address your questions below.
>
>
> **Question A:**
> We use ensembles to improve the calibration[1] of our output probabilities and choose ten as our ensemble size as that is the maximum number of T5-small models that we could fit on our device for decoding.  Please see the performance of different ensemble sizes in the next answer.
>
>
> **Question B:**
> For the Signalling game, we use an RNN with 2 hidden layers and a hidden size of 256. This setup is not sensitive to the choice of hyper-parameters – the results are substantially the same across different learning rates, optimizers, and model sizes.
> For the IGLU building task, we use a pretrained T5-small and fine tune it five times with different random seeds. The choice of seed determines an ordering of the training data. In order to increase diversity in the ensemble, we also trained five models from scratch. We use the AdamW optimizer with a starting learning rate of 1e-5. All models are trained for 10 epochs and we use the checkpoints with the best performance on the validation set. For validation, we use a concatenation of the IGLU-MULTI dataset  and the validation portion of the MDC dataset.
>
> We examine the performance of different ensemble sizes below.
>
> |                                        | IGLU-MULTI avg NLL |
> |----------------------------------------|------------------------|
> | 1 fine-tuned model                     | 0.41                   |
> | 3 fine-tuned models                    | 0.367                  |
> | 5 fine-tuned models                    | 0.354                  |
> | 5 fine-tuned and 5 from scratch models | 0.231                  |
>
>
> We will  include these details on the ensemble member creation in the final manuscript. We will also release code for reproduction and the trained model checkpoints.
>
> **Question C:**
> Both the ambiguous and unambiguous instructions are embedded with a T5-model where we prepend the dialog history and the built state. We have not examined the embedding space itself, only the shape of the output distributions in the two contrasting cases.
>
>
> [1] Simple and Scalable Predictive Uncertainty Estimation using Deep Ensembles. Lakshminarayanan et. al., 2017

---

### Meta-Review · Area_Chair_achv · 2023-10-04

**Recommendation:** 3

**Metareview:**

Improving collaboration using clarification questions and thus reducing uncertainty in contained information. There are some concerns that reviewers raised concerns that have not been addressed, and some reviews have questions about its novelty.

---

### Decision · Program_Chairs · 2023-10-07

**Decision:**

Accept-Findings

**Comment:**

Improving collaboration using clarification questions and thus reducing uncertainty in contained information. There are some concerns that reviewers raised concerns that have not been addressed, and some reviews have questions about its novelty.